# Predicting Antimicrobial and Other Cysteine-Rich Peptides in 1267 Plant Transcriptomes

**DOI:** 10.3390/antibiotics9020060

**Published:** 2020-02-04

**Authors:** Andrey Shelenkov, Anna Slavokhotova, Tatyana Odintsova

**Affiliations:** 1Vavilov Institute of General Genetics, Russian Academy of Sciences, Gubkina str. 3, Moscow 119991, Russia; 2Central Research Institute of Epidemiology, Rospotrebnadzor, Novogireevskaya str. 3a, Moscow 111123, Russia; 3Chumakov Federal Scientific Center for Research and Development of Immune-and-Biological Products of Russian Academy of Sciences, Moscow 108819, Russia

**Keywords:** bioinformatics, antimicrobial peptides, plant transcriptome analysis, motif searching

## Abstract

Antimicrobial peptides (AMPs) are a key component of innate immunity in various organisms including bacteria, insects, mammals, and plants. Their mode of action decreases the probability of developing resistance in pathogenic organisms, which makes them a promising object of study. However, molecular biology methods for searching for AMPs are laborious and expensive, especially for plants. Earlier, we developed a computational pipeline for identifying potential AMPs based on the cysteine motifs they usually possess. Since most motifs are too species-specific, a wide-scale screening of novel data is required to maintain the accuracy of searching algorithms. We have performed a search for potential AMPs in 1267 plant transcriptomes using our pipeline. On average, 50–150 peptides were revealed in each transcriptome. The data was verified by a BLASTp search in nr database to confirm peptide functions and by using random nucleotide sequences to estimate the fraction of erroneous predictions. The datasets obtained will be useful both for molecular biologists investigating AMPs in various organisms and for bioinformaticians developing novel algorithms of motif searching in transcriptomic and genomic sequences. The results obtained will represent a good reference point for future investigations in the fields mentioned above.

## 1. Introduction

Antimicrobial peptides (AMPs) and other cysteine-rich peptides represent one of the major components of innate immunity in various groups of organisms including insects, mammals and plants [1,2]. AMPs are interesting agents for developing novel antimicrobial and antifungal drugs since their unique mode of action greatly decreases the likelihood of resistance development [3]. Studying of plant AMPs, in particular, can elucidate plant-pathogen interactions and provide useful information regarding the innate immunity mechanisms [4]. However, molecular biology-based methods of searching for new AMPs are complex, laborious and expensive.

Earlier we have developed a pipeline [1,5] for fast and accurate screening of RNA-seq data to reveal potential AMPs and other cysteine-rich peptides in various plant transcriptomes. This pipeline is based on so-called “cysteine motifs” representing the arrangement of cysteine residues interleaved by other amino acids in a polypeptide chain. Such motifs are usually developed empirically based on the available AMP data [5,6], so the more data are available, the better the motifs will be in terms of specificity and sensitivity. However, most of the existing motifs were derived just from 10–20 plant transcriptomes or even from model plant data only (e.g., *Arabidopsis thaliana*).

To overcome this existing limitation, we have performed searching for AMPs and cysteine-rich peptides using our pipeline and cysteine motifs currently available in the RNA-seq data obtained from the 1000 plant transcriptomes (1kP) project [7]. This dataset currently contains 1267 transcriptomes from different plant families and thus will provide the comprehensive background for AMP repertoire in plants and will allow us to derive more precise and sensitive/specific motifs for future use by the scientific community. In addition, such studies may facilitate discovering new AMPs that possess stronger antimicrobial activity, e.g., defensins, by dramatically decreasing the number of potential targets to be tested in vitro or in vivo.

## 2. Results

### 2.1. General Description

The current version of Cysmotif searcher includes 138 cysteine motifs divided into the following families: hevein-like peptides, defensins, thionins, non-specific lipid-transfer proteins, snakins, cyclotides, and peptides with unknown function (possibly representing new families with novel functions). Amino acid sequences revealed by SPADA [8], the third-party program included in our pipeline, or those not passing the filtration criteria are assigned to the artificial “cysteine-rich peptides” group. All potential AMPs from the families defined above are cysteine-rich, so this artificial group just indicates that some peptide does not fit to any of these families and does not assume that defensins or thioinins are not cysteine-rich.

Since the number of transcriptomes studied exceeds 1000, the complete results are presented online at the Cysmotif searcher webpage https://github.com/fallandar/cysmotifsearcher (data1k folder) and in Appendix A. These data include the tables showing the numbers of peptides from each family (given above) found in each transcriptome.

Each sample in the 1kP project has a unique four-letter identifier, so we will present these identifiers below together with Latin or common names for the plants to facilitate rapid cross-referencing. In general, the total number of cysteine-rich peptides and AMPs revealed by the pipeline in each transcriptome ranged from 1 (YDCQ, *Cephaleuros virescens*) to 573 (QJYX, *Oltmannsiellopsis viridis*) with a median value equal to 77. Most of these peptides belonged to artificial group, while the number of AMPs ranged from 0 (several species) to 93 (PZAP, *Eleusine coracana*) with a median value equal to 16 (see Figure 1). As we have suggested earlier [5], such a wide range is likely to occur due to differences in transcriptome quality (source material collection and sequencing) rather than biological or molecular characteristics.

Since sequence similarity-based annotation (e.g., using BLAST and corresponding protein databases) remains a common first-line tool for searching the proteins of specific family within a transcriptome, we have performed such annotation for the peptides revealed. In general, BLAST equipped with current versions of databases performs well in predicting snakins and lipid-transfer proteins, but not as well for defensins. As we have shown earlier [1,5,9], our pipeline allows making annotations more specific and accurate, and annotating some peptides with vague motifs which cannot be revealed by homology-based search.

Results for two exemplary transcriptomes (RMVB, *Avena fatua* and CWYJ, *Heracleum lanatum*) are presented below in Table 1 just to make the data description clear. Full data are available online at Cysmotif searcher webpage.

The summary for the total number of peptide groups revealed in the whole transcriptome dataset is given in Table 2. Full data with detailed distribution of AMP groups among plant families and detailed statistics for each transcriptome is presented in Appendix A.

From Table 2, it is easy to see that the most abundant potential AMP group (excluding artificial cysteine-rich one) is snakins. They represent rather large molecules (>7 kDa) found to be active against fungal and bacterial pathogens. Initially, this family was found in the Solanaceae plant family, and the out pipeline has managed to reveal more than 100 snakins in plants from this family. However, the largest number of snakins was found in the Onagraceae (228), Fabaceae (210), and Asteraceae (176) families. The most common motif for snakins was SNA02 (CX{3}CX{3}CX{8}CX{3}CX{2}CCX{2}CX{1}CX{11}CX{1}CX{12}C), responsible for 92% of the motifs revealed.

Here, “C” indicates cysteine residues, X is any residue except cysteine, independently, and numbers in brackets indicate the range for the number of residues. For example, CX{2,4}C will match both “CARC” and “CATFC,” but not “CACTC” (one more cysteine in between) or “CAATRFC” (more than four residues in between terminal cysteines). We will use this notation throughout the manuscript.

The second largest family was defensins. Plant defensins are the best known, and likely most abundant, of all plant AMPs with membranolytic functions, according to data mining of selected plant genomes [10]. In the corresponding section below, we will focus on defensins revealed by our pipeline since this was the only AMP group present in most transcriptomes (>87%) and plant families (>89%) from the dataset considered.

More than 3000 putative AMPs were assigned to lipid-transfer proteins (LTP) by our pipeline. Like snakins, LTPs are rather long cationic proteins (>7 kDa) of approximately 70–90 amino acids with eight cysteine residues. They are distinguished from other AMPs by their lipid transfer activity, in which they bind a wide range of lipids including fatty acids (C10–C14), phospholipids, prostaglandin B2, lyso-derivatives, and acyl-coenzyme A [10]. LTPs were also found in large number of plant families, but they were less abundant than defensins. LTP-rich plant families included Asteraceae (232), Onagraceae (198) and Cupressaceae (130). The most common LTP motif was LTP016: CX{7,9}CX{12,14}CCX{8,19}CX{1}CX{19,23}CX{13,15}C (constituted about 80%).

It is interesting that snakins, defensins and LTPs were revealed by our pipeline in large numbers (>3000) and within the large fraction of the transcriptomes considered, while AMPs from other groups were found in much lower numbers (463 for hevein-like, 412 for thionins, and 30 for cyclotides, respectively). However, this fact could indicate a potential bias caused by plant representation in the dataset or motif stringency for these families.

Hevein-like peptides are basic peptides of 29–45 amino acids with three to five disulfide bonds. They are rich in glycine and contain conserved aromatic residues found in the hevein domain of lectins [11]. Hevein domains bind to chitin [12], which is their primary target. We revealed hevein-like peptides, mostly in plants from the Cupressaceae (68), Portulacaceae (23), and Podocarpaceae (19) families. For these family the most abundant motif was HEVHIP02: CX{3,8}CX{4}CCX{5}CX{6}CX{3,5}CX{1,3}C (about 63%).

The next AMP group is thionins, which represent cationic peptides of 45–48 amino acids with three or four disulfide bonds [13]. Initially, they were known as plant toxins because of their toxicity toward bacteria [10], fungi [14], plant and animal cells [15], and insect larvae. Notably, we found the largest number of thionins in Chlamydomonadaceae family (36), followed by Papaveraceae (16) and Desmidiaceae (10), respectively. In this case, the most common motif was THI018: CCX{11}CX{9,15}CX{5}CX{6,11}C (about 40%).

Finally, cyclotides were revealed mostly in the plants from the Violaceae family (15 of 30), and this is not surprising, since the abundance of this AMP group in Violaceae was previously confirmed [16]. However, cyclotides were also found by our pipeline in Rubiaceae (5), Poaceae (4), and also Papaveraceae, Marattiaceae, Hydrangeaceae, Selaginellaceae, Gentianaceae, and Chrysobalanaceae (one AMP in each family). They are highly variable in sequence, but conserved in structure, and could be divided into two types: Mobius and bracelet. Mobius types contain one *cis*-Pro in loop 5 and a twist in the cyclic backbone, while the bracelet type does not [17]. Cyclotides have potential pharmacological applications, considering their antimicrobial, anti-HIV, anti-tumor, and neurotensin activities. The most common cyclotide motif was CYC02: CX{3}CX{4,5}CX{4,6}CX{1}CX{4,5}C, but it just slightly outperformed another one—CYC01: CX{3}CX{4}CX{6}CX{1}CX{4}C (16 vs. 14 total motifs revealed, respectively).

### 2.2. Defensins

Totally, 4071 sequences possessing nine different motifs assigned to defensins family were revealed in the transcriptomes analyzed. However, more than 90% of the sequences had one of the three motifs with identifiers DEF06 (CX{10}CX{5}CX{3}CX{9}CX{6}CX{1}CX{3}C), DEF32 (CX{4,25}CX{2,12}CX{3,4}CX{3,17}CX{4,32}CXCX{1,6}C) and DEF34 (CX{2,14}CX{3,5}CX{3,16}CX{4,28}CXC).

The largest numbers of defensins belonged to Asteraceae (162), Fabaceae (161), and Poaceae (150) plant families, but this data only reflects the abundance of plants from these families in initial data. If we normalize these values by the number of plants in each family, that is, divide the number of motifs found in any plant family by the number of plant samples belonging to that particular family, we will obtain that the winners are Dunaliellaceae (nine per organism), Chlamydomonadaceae, Poaceae, Portulacaceae, and Solanaceae (seven per organisms for each). Here we have excluded plant families having less than five transcriptomes sequenced to reduce possible bias.

Sequence comparison and multiple alignment can provide the insights into similarities or differences of defensing repertoire from different plant families. It is easy to see that DEF06 is rather precise motif, while DEF32 and DEF34 are more general. Therefore, we will use DEF06 to infer some significant sequence properties of defensins possessing this motif. Alignment of all sequences possessing this motif made by Clustal Omega [18] is shown in Appendix A.

The only fully conserved site except cysteines is glycine in 30th position of the motif (below, all positions described will be referenced starting from the first position of the motif, not the open reading frame or signal peptide). However, if we consider the residues that are conservative in >70% of transcriptomes, 15 more positions are revealed that constitutes about 30% of the motif length. Consensus sequence is presented below.

CESQSHRFKGzCVSzSNCANVCRTEGFzGGzCRGFRRRCFCTKzC

Here, “z” indicates positions, for which consensus includes more than one residue. It is easy to see that this consensus contain mostly charged and hydrophobic residues at the end, while the beginning is mostly polar, and hydrophobic residues are presented in the middle. Most sequences have aromatic amino acids in eighth position of the motif—70% have phenylalanine, the rest include thyrosine and, more rarely, tryptophan. Less than 10 plants from the whole dataset do not have the aromatic amino acid in this position. Other highly conservative amino acids are glycine in 10th position that can be found in all but 10 samples analyzed, and glutamic acid in position 25 revealed in 95% of the samples. Approximately, 4% of plants have aspartic acid in 25th position, and one to three samples include either arginine or glutamine. Interestingly, a large cluster of conservative amino acids (RRRCFCT) is located at the N-terminus of the molecule. It was found in more than 80% of samples, and is likely responsible, among other factors, for interactions with pathogenic microorganisms.

## 3. Discussion

There are several noteworthy aspects of the presented dataset. First, it discloses more than 4000 potential defensins, a family of AMPs with promising anti-bacterial and anti-fungal activity, which are very hard and expensive to be revealed by molecular biology techniques. Second, it includes the results of homology-based annotation of AMPs and other cysteine-rich peptides, which can be used for cross-validation of the data obtained by our pipeline. Third, it includes the cysteine motif data for all AMPs found that will greatly facilitate algorithm training and future development in this field. We plan to use these data for motif refinement in our pipeline.

This dataset can be used for rapid screening of AMP repertoire for a specific plant or plant family in order to reveal the peptides of interest for further investigations. For example, we have already chosen several defensins to check for their anti-fungal activity since they appear to be more potent than currently available peptides.

In addition, we provide the pipeline description and source files, so the users will be able to use it for searching AMPs in their own datasets, for example, containing mammal or insect data, or to make updates as more transcriptomes become available in 1 kP project.

The availability of AMP data for more than 1000 different plant transcriptomes will be useful for future studies in this field, including algorithm training, motif mining, and studying cysteine-rich peptide repertoire of particular plant families or other groups of species. Together with the homology search data provided, this information will greatly facilitate various investigations in plant proteomics and allow the narrowing the molecular biology experiments to save working time and resources.

## 4. Materials and Methods

### 4.1. Source Data

Initial transcriptomic data in FASTA format for 1267 plant transcriptomes available on 15 April 2019 were downloaded from 1kP website [7]. Cysmotif searcher pipeline [5] version 3.2 was used for searching AMPs in all transcriptomes above; default parameters and –k 5 –s -l 150 options were used. The options included calling SPADA software [8] (last update: 6 June 2017) integrated within Cysmotif searcher. SignalP 4.1 [19] was used for signal peptide prediction that represents an important stage of pipeline filtering process.

### 4.2. Bioinformatics Tools

A homology search for the final output results was performed using local BLAST 2.2.31+ (blastp) for nr and SwissProt [20] databases (retrieved at March 21, 2019).

Cysmotif searcher pipeline was described earlier [5]. Generally, it performs cysteine motif searching in translated nucleotide sequences using regular expressions and then subjects the motifs found to a number of filtration steps to confirm that they are not artifacts. Filtration includes checking for the presence of open reading frame starting with methionine, the presence of signal peptide (using SignalP [19]) and length restriction (less than 150 in the current case). SPADA [8] can be also called within pipeline to search for additional cysteine-rich peptides which does not include the motifs under study but may represent interesting additional targets for future investigations. The cysmotif searcher pipeline was also tested on randomly generated amino acid sequences to verify the validity of its results [5]. The source code is available on github at https://github.com/fallandar/cysmotifsearcher. Version 3.2 was used to obtain the data for the current manuscript.

For the sake of clarity, the flowchart describing the procedure of potential AMP identification is presented in the Figure 2.

## 5. Conclusions

In this paper we have provided the results of searching AMP and other cysteine-rich peptides in the dataset of 1kP project containing more than 1000 plant transcriptomes from various species. The pipeline developed by us allowed to reveal 50–100 AMPs in each sample, some of which cannot be found by performing homologous search in corresponding databases. We provide comprehensive output data including amino acids motifs found, AMP distribution across plant families and organisms, BLAST search results for the peptides revealed, amino acid sequence alignment etc. We believe that these results will be useful both for molecular biologists, since they can help to reduce the number of experiments required to identify AMPs, and for bioinformaticians working in this field since they allow to refine AMP searching performed by other software and to define new searching models for future use.

## Figures and Tables

**Figure 1 antibiotics-09-00060-f001:**
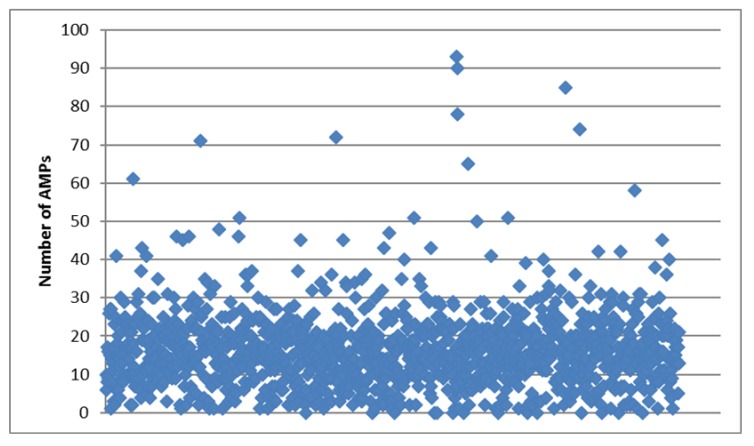
Scatter plot for the number of antimicrobial peptides (AMPs) revealed in the transcriptomes analyzed.

**Figure 2 antibiotics-09-00060-f002:**
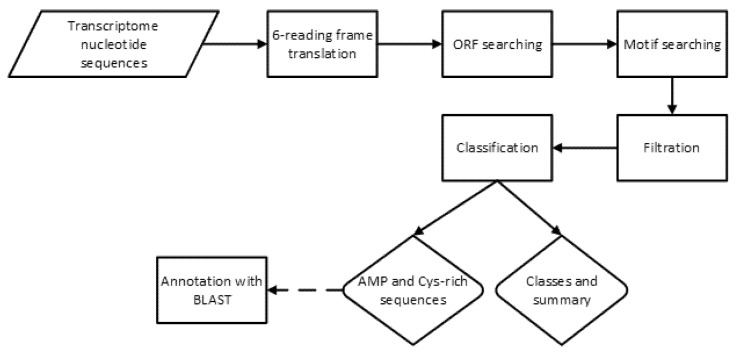
Flowchart of AMP identification process using a Cysmotif searcher pipeline.

**Table 1 antibiotics-09-00060-t001:** Example distributions of potential AMPs among peptide families.

Organism/Family	Defensins	Thionins	Cyclotides	Snakins	Hevein-Like	LTP	Cysteine-Rich	Unknown	No BLAST Hits
*A. fatua*	5/31/3 *	0/0/0	0/0/0	1/2/0	0/0/0	12/4/8	7/18/5	15/1/12	6/0/3
*H. lanatum*	5/22/3	3/0/1	0/0/0	25/12/13	0/0/0	37/5/32	35/29/30	11/0/10	27/0/17

* N1/N2/N3 represents results for the number of peptides revealed by BLAST annotation only (without using our pipeline)/Cysmotif searcher classification/SPADA + BLAST, respectively.

**Table 2 antibiotics-09-00060-t002:** Total number of motifs revealed for each peptide group in 1267 transcriptomes.

AMP Group	Total Number of Motifs
Cysteine-rich (artificial group)	32,102
Snakins	4849
Defensins	4071
Lipid-transfer proteins	3338
Potential new families of AMP	847
Hevein-like peptides	463
Thionins	412
Cyclotides	30

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
