# Peer review of "Predicting Antimicrobial and Other Cysteine-Rich Peptides in 1267 Plant Transcriptomes"

_antibiotics, 2020, doi:10.3390/antibiotics9020060_

Round 1
Reviewer 1 Report
A nice pictorial flow chart for the procedure to identify peptides from the tool is required.
The authors may take a few of these peptides and randomly screen for their antimicrobial activity in vitro.
Author Response
1. A nice pictorial flow chart for the procedure to identify peptides from the tool is required.
A graph has been added to the manuscript as Figure 2 ('Bioinformatic tools' section of 'Materials and Methods').
2. The authors may take a few of these peptides and randomly screen for their antimicrobial activity in vitro.
This is very useful suggestion. However, this task requires extensive time- and cost-consuming work since the peptides to be screened should be obtained either by oligonucleotide merging and subsequent expression or by direct amino acid chemical synthesis. In addition, proper folding and disulfide bonds' formation can not be easily achieved in all cases. Generally speaking, such experiments lie beyond the scope of the manuscript since it is dedicated to bioinformatics tool application. In contrast, we believe that the data presented in our manuscript and its online supplements will allow other research groups to perform such a screening in efficient and guided manner.
Reviewer 2 Report
Prof. Shelenkov and colleagues reported a comprehensive search for antimicrobial peptides (AMP) using bioinformatic pipeline. AMPs are important substance in a variety of species, including bacteria, insects, plants, and mammals regarding innate immunity. However, traditional methods (experimental) are less efficient to discover novel AMPs. Standing on the computational pipeline they developed previously, the team applied the methods to a broader screening of data based on the common cysteine-containing motif. Their algorithm was proved to be sustainable. A handful of peptides were identified in each transcriptome. The results could be verified using orthogonal methods. This study lays foundation to researchers who are interested in AMPs performing experimental and computational research. The manuscript was written well. The experiment was designed properly. This reviewer recommends acceptance in current form.
Author Response
No response required since the reviewer recommends acceptance in current form.
Reviewer 3 Report
This is a well written manuscript which may be published in its actual form.
Author Response

(The authors gave the same response as above.)
